# Exercise Training Differentially Affects Skeletal Muscle Mitochondria in Rats with Inherited High or Low Exercise Capacity

**DOI:** 10.3390/cells13050393

**Published:** 2024-02-24

**Authors:** Estelle Heyne, Susanne Zeeb, Celina Junker, Andreas Petzinna, Andrea Schrepper, Torsten Doenst, Lauren G. Koch, Steven L. Britton, Michael Schwarzer

**Affiliations:** 1Department of Cardiothoracic Surgery, Jena University Hospital, Friedrich Schiller University of Jena, 07747 Jena, Germany; estelle.heyne@med.uni-jena.de (E.H.); petzinna@web.de (A.P.); andrea.schrepper@med.uni-jena.de (A.S.); doenst@med.uni-jena.de (T.D.); 2Department of Physiology and Pharmacology, College of Medicine and Life Sciences, The University Toledo, Toledo, OH 43606, USA; lauren.koch2@utoledo.edu; 3Department of Anesthesiology, University of Michigan Medical School, Ann Arbor, MI 48109, USA; brittons@umich.edu

**Keywords:** inherited exercise capacity, exercise training, skeletal muscle mitochondria

## Abstract

Exercise capacity has been related to morbidity and mortality. It consists of an inherited and an acquired part and is dependent on mitochondrial function. We assessed skeletal muscle mitochondrial function in rats with divergent inherited exercise capacity and analyzed the effect of exercise training. Female high (HCR)- and low (LCR)-capacity runners were trained with individually adapted high-intensity intervals or kept sedentary. Interfibrillar (IFM) and subsarcolemmal (SSM) mitochondria from gastrocnemius muscle were isolated and functionally assessed (age: 15 weeks). Sedentary HCR presented with higher exercise capacity than LCR paralleled by higher citrate synthase activity and IFM respiratory capacity in skeletal muscle of HCR. Exercise training increased exercise capacity in both HCR and LCR, but this was more pronounced in LCR. In addition, exercise increased skeletal muscle mitochondrial mass more in LCR. Instead, maximal respiratory capacity was increased following exercise in HCRs’ IFM only. The results suggest that differences in skeletal muscle mitochondrial subpopulations are mainly inherited. Exercise training resulted in different mitochondrial adaptations and in higher trainability of LCR. HCR primarily increased skeletal muscle mitochondrial quality while LCR increased mitochondrial quantity in response to exercise training, suggesting that inherited aerobic exercise capacity differentially affects the mitochondrial response to exercise training.

## 1. Introduction

Aerobic exercise capacity is a strong indicator of mortality not only in patients but also in healthy humans [1,2,3]. Currently, a paradoxical discussion is ongoing, because this association is with respect to atherosclerosis contradicted by studies questioning the beneficial effects of increasing one’s exercise capacity by endurance training [4]. Low exercise capacity is associated with higher morbidity and mortality and with a higher risk for reduced insulin sensitivity [5] as well as cardiovascular disease [6]. Several mechanisms may be responsible for those associations but one of the most relevant is the connection to metabolism. Mitochondria as main producers of adenosine triphosphate (ATP) are important regulators of exercise capacity and there is a clear correlation between high exercise capacity and high mitochondrial function [7].

The largest part of an individual’s exercise capacity, namely 50–70%, is inherited (intrinsic) [8], while the remaining part is dependent on environmental conditions and lifestyle (extrinsic). The contribution of intrinsic and extrinsic factors to exercise capacity in humans is not clear. However, Koch and Britton developed a model of rats with contrasting high (HCR) and low (LCR) inherited running capacity. Rats were selectively bred for these different phenotypes over generations, based on treadmill testing [9]. On the one hand, HCR present with an athletic phenotype and have a significantly longer life expectancy than LCR [10]. On the other hand, LCR have a higher risk of developing diseases and show all signs of metabolic syndrome [11]. Moreover, mitochondrial capacity is lower in the skeletal muscle of LCR compared to HCR [12], highlighting the link between inherited exercise capacity and mitochondrial function.

Skeletal muscle mammalian mitochondria account for about 2–10% of the cell volume, depending on the muscle fiber type [13]. In skeletal muscle, several mitochondrial subtypes exist [14], which include the most prominent interfibrillar (IFM) and subsarcolemmal (SSM) mitochondria. While IFM are located between myofibrils, SSM reside between myofibrils and the sarcolemma. Mitochondria are able to build network functions and the subpopulations can present with different shapes and structures, which again are dependent on muscle fiber type, as well as with different biochemical properties and functions [14]. Different pathologies such as diabetes or calcium-dependent metabolic stress have been shown to differentially affect mitochondrial subpopulations [15,16]. To date, it is not clear if IFM and SSM in skeletal muscle of HCR and LCR are (functionally) comparable.

The most effective way to increase physical performance is exercise training. The effects on health status or survival but also on skeletal muscle properties, for example, muscle gain, depend on the exercise program, duration, as well as intensity [17,18]. Furthermore, in humans, it has been shown that both sex and current training status may change the adaptation to exercise training [19]. Since exercise affects skeletal muscle metabolism, mitochondria are thought to be the key players in eliciting the effects of exercise training by improving their function [7]. One of the discussed mechanisms includes the activation of peroxisome proliferator-activated receptor gamma coactivator 1-alpha (PGC-1α), a transcription factor controlling mitochondrial biogenesis and thus mitochondrial content. Increases in skeletal muscle mitochondrial quality (e.g., respiratory capacity) and quantity (mitochondrial content) have been shown to depend on exercise intensity and volume [20,21]. Succinate dehydrogenase activity increased in human skeletal muscle in IFM linearly throughout combined strength and endurance training while SSM mainly increased late in training [22]. The interplay between intrinsic and extrinsic exercise capacity on skeletal muscle mitochondrial function, however, is still a matter of debate.

We thus aimed to assess mitochondrial function in skeletal muscle subpopulations of rats with high or low inherited exercise capacity as well as the effect of four weeks of high-intensity interval training on a treadmill. We hypothesized that sedentary HCR and LCR present differences in respiratory capacity in IFM and SSM. Furthermore, we expected that exercise training increases exercise capacity of both HCR and LCR as well as mitochondrial content and respiratory capacity.

## 2. Materials and Methods

### 2.1. Animals

All animal procedures were approved by the responsible local authorities (Thüringer Landesamt für Verbraucherschutz) and registered as 22-2684-04-02-112/14 and 22-2684-04-02-082/14. Animals were handled and housed in accordance with the National Institutes of Health (NIH) guidelines. The high- and low-capacity runner (HCR and LCR) rat model was generated by special breeding using a genetically heterogeneous N:NIH stock of rats as the founder population and a velocity-ramped running protocol as previously described [9]. Female HCR and LCR rats from generations 33 and 36 (2 cohorts) were included in this study and used for different investigations (HCR n = 43, LCR n = 38). Cohort 1 included rats from generation 33 and was used for basal characterization and mitochondrial assessments. Cohort 2 included rats from generation 36 and was used for determining exercise capacity pre/post exercise training as well as for an echocardiographic analysis and glucose/insulin tolerance tests. The exact numbers of animals in the different experiments are indicated in the figures/tables accordingly as well as described in the respective method parts. Rats were fed ad libitum and housed at 21 °C with a light cycle of 12 h.

### 2.2. Echocardiography

Echocardiographic examination was performed in HCR and LCR at ~15 weeks of age, as previously described [16]. A total of 32 animals were set up for the experiment (HCR n = 8, HCR ex n = 6, LCR n = 8, LCR ex n = 10). Briefly, animals were anesthetized with isoflurane (1.5%). Chests were shaved and the rats were examined in supine position with a 17.5 MHz phased array transducer (RMV716, VisualSonics, Amsterdam, Netherlands). Two-dimensional short-axis views of the left ventricle at the papillary muscle level were obtained. Two-dimensional-guided M-mode tracings were recorded with a sweep speed of 100 mm/s. We determined left ventricular wall thickness (posterior wall thickness) and cavity size in diastole (left ventricular end-diastolic dimension) by the American Society for Echocardiology leading edge method and averaged values from five measurements for each examination.

### 2.3. Assessment of Aerobic Exercise Capacity

Maximal exercise capacity was performed by running to exhaustion on a treadmill at a 25° incline according to Hoydal et al. in HCR and LCR at the age of ~11 weeks as well as following four weeks of exercise training at the age of ~15 weeks (HCR pre/post exercise n = 6, LCR pre/post exercise n = 9) [23]. After 15 min of adaptation to the treadmill (HCR [0.22 m/s] and LCR [0.07 m/s]), speed was increased stepwise every 2 min by 1.8 m/min, until the animals were unable to run further. The test was repeated twice with a one-day break in-between. Maximal anaerobic exercise capacity (100%) was calculated by subtracting the last 4 steps (7.2 m/min) of maximal speed achieved to account for anaerobic exercise.

### 2.4. Glucose and Insulin Tolerance Test

Rats at ~10 weeks of age were fasted for 6 h before glucose or insulin tolerance tests were performed as previously described [16] (glucose tolerance test: HCR n = 7, HCR ex n = 5, LCR n = 7, and LCR ex n = 8, and in HCR ex, 1 animal had to be excluded due to diarrhea after glucose injection; insulin tolerance test: HCR n = 8, HCR ex n = 5, LCR n = 8, and LCR ex n = 9, and fasting blood glucose data were obtained during both glucose and insulin tolerance test and the results were combined: HCR n = 15, HCR ex n = 11, LCR n = 15, LCR ex n = 17). Tests were performed on separate days with at least one day in-between. Rats were anesthetized with isoflurane (1.5%) for the injection of glucose or insulin as well as for blood sampling. The glucose tolerance test was performed using a single dose of 20% glucose (2 g/kg). The insulin tolerance test was initiated with a single dose of insulin (1 I.U./kg) administered by intraperitoneal injection. One drop of peripheral blood was used at 0, 15, 30, 60, and 120 min to measure blood glucose (mmol/L) using a glucometer (Freestyle mini, Abbott; Wiesbaden, Germany). Area under the curve [24] was calculated for data analyses. 

### 2.5. Aerobic Interval Training

HCR and LCR rats at the age of ~11 weeks were assigned to one of four groups: HCR sedentary (HCR), HCR trained (HCR ex), LCR sedentary (LCR), and LCR trained (LCR ex). Two cohorts of rats were used for this analysis. Cohort 1 rats were from generation 33. They had been trained (HCR n = 11, LCR n = 7; all animals completed the 4 weeks of exercise training; sedentary rats: HCR n = 15, LCR n = 12). For cohort 2, rats from generation 36 had been trained (HCR n = 9, LCR n = 11; completion of the exercise training: HCR n = 6, LCR n = 10; sedentary rats: HCR n = 8, LCR n = 8). Exercise was performed as aerobic interval training (AIT) for 105 min/day, 5 days/week, 4 weeks in total, on a treadmill at a 25° incline as described elsewhere [25]. Two treadmills were used in parallel, allowing a total of 6 rats to be trained. Exercise training was completed in a time window starting at about 7.00 a.m. until the early afternoon. After 15 min of warm-up, animals were trained for 1.5 h in AIT with 8 min at 85% and 2 min at 55% of maximal aerobic exercise capacity. Treadmill velocity was increased by 0.02 m/s weekly if the animal was able to successfully complete at least three out of the five training sessions during one week. Running speed and time were recorded during each session. Running distance was calculated from treadmill speed and running time.

### 2.6. Isolation of Mitochondria

Animals were euthanized, and organs and tissues were harvested at ~15 weeks of age (HCR n = 15, HCR ex n = 11, LCR n = 12, LCR ex n = 7). Skeletal muscle interfibrillar (IFM) and subsarcolemmal (SSM) mitochondria were isolated from freshly explanted gastrocnemius muscle as described before [16,26] with a modified Chappell–Perry buffer (containing 100 mM KCl, 50 mM MOPS, 1 mM EGTA, 5 mM MgSO_4_-7H_2_O, and 1 mM ATP, pH 7.4, 4 °C). IFM were harvested after treating the homogenate with 5 mg of trypsin/g of wet weight of skeletal muscle for 10 min at 4 °C. Freshly isolated mitochondria from gastrocnemius muscle were kept in a KME buffer (100 mM KCl, 50 mM MOPS, 0.5 mM EGTA, at pH 7.4). Mitochondrial protein content was determined by the Bradford method using bovine serum albumin as a standard. Citrate synthase activity was measured in the fresh skeletal muscle homogenate as a marker for mitochondrial mass according to the protocol of Srere et al. [27] (HCR n = 13, HCR ex n = 11, LCR n = 9, LCR ex n = 6).

### 2.7. Mitochondrial Morphometry

Mitochondria were analyzed using flow cytometry FC500 (Beckman Coulter, Krefeld, Germany) according to the method of Dabkowski et al. [28]. In brief, freshly isolated mitochondria were diluted 1/1000 in the KME buffer (100 mM KCl, 50 mM MOPS, 0.5 mM EGTA, at pH 7.4) (IFM: HCR n = 13, HCR ex n = 10, LCR n = 10, LCR ex n = 7; SSM: HCR n= 13, HCR ex n = 9, LCR n = 10, LCR ex n = 6). Forward scatter was used to determine size and sideward scatter was used to determine complexity of mitochondria. Mitochondrial complexity assesses the internal structure (internal membranes) of mitochondria. Higher complexity means higher sideward reflection. The measurement was stopped after analyzing 10,000 mitochondria or after a maximum of 150 s.

### 2.8. Assessment of Mitochondrial Respiratory Rates

The oxygen consumption of isolated mitochondria was measured using a Clark-type oxygen electrode (Strathkelvin, North Lanarkshire, Scotland) at 25 °C as described elsewhere [16]. Mitochondria were incubated in a solution consisting of 100 mM KCl, 50 mM MOPS, 1 mM EGTA, 5 mM KH_2_PO_4_, and 1 mg/mL of fatty-acid-free bovine serum albumin at pH 7.4. The rate of oxidative phosphorylation was measured using 10 mM glutamate, 10 mM glutamate/2.5 mM malate, 20 µM palmitoylcarnitine/2.5 mM malate, 10 µM palmitoyl-CoA/2.5 mM malate/0.5 mM carnitine, 5 mM pyruvate/2.5 mM malate, 10 mM succinate/3.75 µM rotenone, 0.5 mM durohydroquinone/3.75 µM rotenone, or 0.5 mM N,N,N′,N′-Tetramethyl-p-phenylendiamine (TMPD)/5 mM ascorbate/3.75 µM rotenone as substrates and ADP as a stimulus. The ADP-stimulated oxygen consumption (state 3) and the ADP-limited oxygen consumption (state 4) in the respiratory chamber as well as the ADP/O ratio (ADP added per oxygen consumed) were determined. The respiratory control ratio (RCR) was calculated as state 3 respiration to state 4 respiration. To determine uncoupled respiration, state 3 respiration was measured in the presence of 0.1 mM dinitrophenol (DNP). Material from 43 animals (HCR n = 13, HCR ex n = 11, LCR n = 12, LCR ex n = 7) was used for assessments. Due to the limited amount of isolated mitochondria or outliers (calculated by Grubbs test), the group sizes may differ in mitochondrial subpopulations depending on the substrate used (IFM: HCR n = 4–13, HCR ex n = 7–11, LCR n = 5–12, LCR ex n = 6–7; SSM: HCR n = 5–12, HCR ex n = 6–11, LCR n = 5–11, LCR ex n = 3–6).

### 2.9. Determination of Isolated Complex Activities

Mitochondrial isolated complex activities were measured as described before [16]. Freshly isolated mitochondria were treated with 1 mg of cholate/mg of mitochondrial protein. After one cycle of freeze/thaw (−80 °C–+25 °C), electron transport chain (ETC) complex activities were measured as specific donor–acceptor oxidoreductase activities. Complex I was measured as the rotenone-sensitive reduction of 2,6-dichloroindophenol (DCIP) with NADH as a substrate. The reduction of DCIP with succinate as a substrate assesses complex II. Complex III activity was determined as the antimycin-A-sensitive reduction of cytochrome c, using decylubiquinol as a substrate. Cytochrome c oxidase (complex IV) was measured as the oxidation of reduced cytochrome c. The combined activities of complexes I + III and complexes II + III were determined as described before [29]. Isolated mitochondria of 43 animals were assessed. The group size of animals differed in the individual complex measurements due to a limited amount of isolated mitochondria or outliers (calculated by Grubbs test): (IFM: HCR n = 11–15, HCR ex n = 10–11, LCR n = 4–11, LCR ex n = 4–7; SSM: HCR n = 11–14, HCR ex n = 9–10, LCR n = 4–5, LCR ex n = 6).

### 2.10. Statistical Analysis

For the preparation of figures and the statistical analysis, Sigmaplot and Sigmastat were used. Data are presented as the mean ± SEM and were analyzed using two-way ANOVA for comparing all datasets except for the exercise analysis of exercise capacity pre and post exercise training (two-way repeated measurement ANOVA). Post hoc comparisons among the groups were performed using the Bonferroni method. Differences among groups were considered statistically significant if *p* < 0.05, and main effects for the HCR/LCR population and treatments are given in figures and tables. Outliers were detected by using Grubbs‘ test via a GraphPad calculator (https://www.graphpad.com/quickcalcs/grubbs1/; accessed on 20 Ferbruary2023). Sample size was calculated using an online calculator (http://powerandsamplesize.com/Calculators/Compare-2-Means/2-Sample-Equality; accessed on 12 May 2014) according to Chow et al. [30].

## 3. Results

We assessed the effects of exercise training on exercise capacity, muscle weight (Figure 1), and glucose handling (Figure 2). Animals were tested for exercise capacity at the age of 11 weeks and again 4 weeks later after the exercise training period. In sedentary LCR, the best treadmill running speed (Figure 1a) was about 60% lower than HCR and LCR could only run for half the time (Figure 1b) of HCR until exhausted. These findings confirm the previously described genetically determined differences in exercise capacity between HCR and LCR. Four weeks of exercise training led to an increase in running speed (Figure 1a; HCR: *p* = 0.03, LCR: *p* < 0.001) as well as time to exhaustion (Figure 1b; HCR: *p* = 0.030, LCR: *p* < 0.001) in both HCR and LCR. This effect was stronger in LCR and thus exercise training attenuated the difference in exercise capacity between HCR and LCR. Skeletal muscle weights of gastrocnemius muscle (Figure 1c) and soleus muscle (Figure 1d) were comparable between sedentary HCR and LCR in relation to body weight. Exercise training resulted in a significantly higher weight of soleus in both HCR and LCR, showing that exercise training was efficient in both lines (Figure 1d; HCR: *p* < 0.001, LCR: *p* = 0.743, exercise effect: *p* < 0.001). Instead, gastrocnemius weight was not affected by exercise training, indicating that the hypertrophic response depended on the muscle fiber type. Exercise training induced phenotypic changes in both HCR and LCR but seemed to be more effective in LCR, suggesting a higher trainability.

Glucose and insulin tolerance (GTT/ITT) were tested at the age of 14 weeks in sedentary and trained HCR and LCR (Figure 2a–e). Fasting blood glucose (Figure 2a) was higher in sedentary LCR compared to HCR (*p* = 0.010). Moreover, GTT revealed a higher area under the curve (AUC) (Figure 2c, *p* = 0.001) and a delay in the response (Figure 2b) for sedentary LCR compared to HCR, indicating decreased glucose tolerance. Exercise training had opposite effects in HCR and LCR with a slight increase in AUC in HCR (*p* = 0.221) and a slight decrease in AUC in LCR (*p* = 0.054), resulting in comparable glucose tolerance between trained HCR and LCR (Figure 2c). Insulin tolerance (Figure 2d,e) showed a more pronounced difference between sedentary HCR and LCR. The area under the curve (Figure 2e) was about 45% higher in LCR compared to HCR and the difference between both was even increased after exercise training (HCR ex vs. LCR ex: *p* = 0.002), suggesting impaired insulin tolerance in trained and untrained LCR (Figure 2d,e). Glucose metabolism and insulin tolerance were impaired in sedentary LCR but the differences were lost in exercise-trained animals.

Table 1 shows animal morphometric characteristics and results of cardiac functional assessment (additional data are shown in Appendix A) by echocardiography of sedentary and trained HCR and LCR. As expected, body weights and liver weights of sedentary LCR were significantly higher compared to HCR (body weight: *p* < 0.001, liver weight: *p* < 0.008) while heart and lung weights in relation to tibia length were comparable between both lines. In addition, sedentary LCR presented with higher fractional shortening (*p* = 0.001), indicating higher systolic function. Four weeks of exercise training increased body weight in HCR (*p* = 0.003) but did not affect body weight in LCR (*p* = 0.366). In parallel, lung and liver weights related to tibia length increased with exercise training in HCR (lung: *p* = 0.002, liver: *p* < 0.001). In the heart, the increase in weight due to exercise training was more pronounced in HCR (HCR: *p* < 0.001, LCR: *p* = 0.342), indicating more pronounced cardiac hypertrophy. Consistently, hearts of exercised HCR showed a greater increase in left ventricular internal diameter in diastole (LVIDd) compared to trained LCR (HCR: *p* = 0.053, LCR: *p* = 0.158). Thus, cardiac hypertrophy seemed to be more prominent in HCR.

Skeletal muscle mitochondria are the main support for ATP, and their amount and morphology were investigated and the results are shown in Figure 3. Citrate synthase activity of muscle tissue homogenate (Figure 3a) as a measure for mitochondrial mass was more than twice as high in sedentary HCR compared to LCR (*p* < 0.001). Exercise induced an increase in mitochondrial mass, which was more pronounced in LCR (70% increase, *p* = 0.088) than in HCR (18% increase, *p* = 0.068), indicating an improvement in mitochondrial quantity in both HCR and LCR (exercise effect: *p* = 0.016). Nevertheless, citrate synthase activity remained significantly higher in trained HCR than in LCR (*p* = 0.005). We isolated and separated interfibrillar (IFM) and subsarcolemmal (SSM) mitochondria of gastrocnemius muscle and determined mitochondrial morphology. Mitochondrial size of IFM (Figure 3b) was not different between sedentary HCR and LCR but was reduced with exercise in both (HCR ex: *p* = 0.069, LCR ex: *p* = 0.085). In contrast, mitochondrial complexity in IFM (Figure 3c) was higher in LCR compared to HCR independent of exercise training (exercise effect: *p* = 0.005). In SSM, size (Figure 3d) and complexity (Figure 3e) were affected by neither genotype nor by exercise training. Mitochondrial quantity was higher in sedentary HCR and increased with exercise especially in LCR.

Mitochondrial maximal respiratory capacity indicating mitochondrial function is shown in Figure 4 for IFM. Respiratory capacity was significantly higher in sedentary HCR than in sedentary LCR for complex I substrates glutamate/malate (Figure 4b, *p* = 0.010) and pyruvate/malate (Figure 4c, *p* = 0.012) as well as complex IV substrate TMPD + ascorbate (Figure 4h, *p* = 0.036). In addition, there was a trend for higher respiratory capacity in sedentary HCR with complex I substrate palmitoylcarnitine/malate (Figure 4e, *p* = 0.097) and with complex II substrate succinate (Figure 4f, *p* = 0.064). Exercise training affected maximal respiratory capacity in HCR but not in LCR. Maximal respiratory capacity was significantly increased in trained HCR using complex III substrate DHQ (Figure 4g, *p* = 0.031). With substrates glutamate (Figure 4a), palmitoyl coenzyme A/carnitine/malate (Figure 4d), palmitoylcarnitine/malate (Figure 4e, *p* = 0.285), or succinate (Figure 4f, *p* = 0.209), maximal respiratory capacity was also higher in exercised compared to sedentary HCR but did not reach statistical significance. The difference in respiratory capacity in *gastrocnemius* IFM of sedentary HCR and LCR was increased with exercise training, resulting in higher respiratory capacity of trained HCR compared to trained LCR. The differences reached statistical significance with all substrates used except for glutamate and palmitoyl-carnitine/malate. This suggests an important impact of exercise training on mitochondrial function in HCR. Appendix A shows additional parameters characterizing mitochondrial function of gastrocnemius IFM. Four weeks of exercise significantly increased the ADP/O ratio in IFM of HCR with three substrates and of LCR with four substrates. The overall effect of exercise (Ex) on both lines showed statistical significance for six out of seven substrates, indicating a higher efficiency (ADP/O) of respiratory chain function in both genotypes. The ADP-limited respiration (state 4), respiratory control index (RCI), and uncoupled respiration (DNP) mainly indicated differences between HCR and LCR but not with exercise training. This suggests that these factors are irrelevant for differences between sedentary and trained rats. Interfibrillar mitochondrial function was thus higher in sedentary HCR and increased with exercise in HCR.

Furthermore, we measured maximal respiratory capacity in SSM of gastrocnemius (Figure 5). SSM showed no prominent inherited differences in respiratory capacity between HCR and LCR (Figure 5a–h). Four weeks of exercise training did not affect respiratory capacity in SSM of HCR or LCR. Appendix A shows additional parameters further characterizing SSM function of gastrocnemius. Respiratory chain efficiency estimated by the ADP/O ratio was significantly higher in HCR compared to LCR with fatty acid substrates (palmitoyl coenzyme A; factor inherited exercise capacity: *p* = 0.041, palmitoylcarnitine: *p* = 0.044). Following exercise training, respiratory chain efficiency increased in both HCR and LCR and showed statistical significance with the substrates glutamate/malate (exercise effect: *p* = 0.020), pyruvate/malate (exercise effect: *p* = 0.012), and succinate/rotenone (exercise effect: *p* = 0.014). Here, the ADP-limited respiration (state 4), respiratory control index (RCI), and uncoupled respiration (DNP) only rarely indicated differences between HCR and LCR and no differences with exercise training. Thus, these factors seem irrelevant for differences with intrinsic exercise capacity or exercise training in our rats.

We assessed isolated complex activities in IFM of gastrocnemius muscle (Figure 6a–g) to investigate if the differences in respiratory capacity in HCR and LCR can be explained by differences in complex activities. In IFM, we found significantly higher activities in sedentary HCR compared to LCR with complex II–III (Figure 6h, *p* = 0.037) as well as complex IV (Figure 6d, *p* = 0.009). Higher activities of complex I (Figure 6a, *p* = 0.168), III (Figure 6c, *p* = 0.206), and I–III (Figure 6g, *p* = 0.168) were also observed in sedentary HCR compared to sedentary LCR but did not reach statistical significance. The factor inherited exercise capacity (G) was significantly different with all complexes except for complex II and V, indicating an effect of genetics on complex activities. Exercise training increased isolated complex activities in HCR but not in LCR. This increase was significant with complex II (Figure 6b, *p* = 0.041), III (Figure 6c, *p* = 0.009), I–III (Figure 6g, *p* < 0.001), and IV (Figure 6d, *p* = 0.032). These data suggest that the increase in maximal respiratory capacity in IFM of HCR following exercise seems at least to be in part due to an increase in isolated complex activities. In contrast, isolated complex activities of SSM (Figure 7a–g) showed a different pattern compared to IFM. Differences between sedentary HCR and LCR were somehow similar in IFM and showed several higher complex activities in HCR but without reaching statistical significance. A significant response to four weeks of exercise was found in SSM of HCR with complex I (Figure 7a, *p* = 0.002), II (Figure 7b, *p* = 0.015), III (Figure 7c, *p* = 0.004), I–III (Figure 7g, *p* < 0.001), II–III (Figure 7h, *p* = 0.013), and IV (Figure 7d, *p* = 0.043). Furthermore, the exercise training effect (Ex) was significantly different with all complexes except for complex V, indicating an effect of exercise training on complex activities in both HCR and LCR. The observed differences in SSM complex activities between HCR and LCR with or without exercise training were not reflected in respiratory capacity in SSM of gastrocnemius. Thus, isolated complex activities may explain findings in respiratory capacity of IFM but not SSM.

## 4. Discussion

In the present investigation, we showed that high intrinsic exercise capacity is associated with increased skeletal muscle mitochondrial mass compared to low intrinsic exercise capacity. However, the increase in mitochondrial mass in response to exercise was more pronounced in LCR than in HCR. Mitochondrial function was higher in gastrocnemius muscle interfibrillar mitochondria in sedentary HCR compared to LCR. Exercise training increased mitochondrial respiration in skeletal muscle IFM of HCR only. In contrast, subsarcolemmal mitochondria seem to not be involved in the differences between low and high intrinsic exercise capacity or in the response to exercise training. We here show for the first time that differences in the function of mitochondrial subpopulations are mainly related to the inherited part of exercise capacity. Exercise training increased exercise capacity more in LCR, which altogether suggests that inherited high or low exercise capacity differentially affects both the gross physiological and mitochondrial response to exercise training.

High exercise capacity has been described to prevent morbidity and mortality. However, lifelong endurance athletes present with a higher abundance of coronary plaques when compared to healthy humans [4], questioning the effect of endurance training. Thus, it is conceivable that genetic factors have been underestimated in this context. We compared two rat strains with diverging inherited exercise capacity and our results clearly show higher respiratory capacity in skeletal muscle IFM with HCR compared to LCR but no differences in SSM. Differences in mitochondrial respiratory capacity in gastrocnemius muscle have been described before for isolated mitochondria containing mixed subpopulations [31,32] or skinned soleus muscle fibers of HCR and LCR [33]. These reports consistently showed higher respiratory capacity in HCR compared to LCR. However, no analysis discriminating between the two mitochondrial subpopulations has been performed in animals with different intrinsic exercise capacities. Thus, the presence of mitochondrial functional differences with high or low intrinsic exercise capacity depends on the mitochondrial subpopulation. Depending on the rat model and substrate used, higher [16] or no difference [34] in mitochondrial respiratory capacity has been described for IFM compared to SSM in gastrocnemius muscle, but differences due to contrasting intrinsic exercise capacity are new. It has been suggested that IFM and SSM are relevant for different cellular functions. While IFM may be responsible for providing mainly energy for contraction [35,36], SSM have shown to be implicated mainly in cell maintenance and homeostasis of calcium and may be more protected from calcium overload than IFM [37]. Our data suggest that increased interfibrillar mitochondrial function may be responsible for increased muscle capacity with high compared to low inherited exercise capacity supporting the hypotheses that IFM function is relevant for energy supply to skeletal muscle and that the genetic constitution mainly determines health and skeletal mitochondrial subpopulation function.

Endurance exercise training can be performed at different intensities in combination with different frequencies. Ineffective exercise has been suggested as a cause for a lack of measurable exercise effects [38]. In athletes and in patients, relative protocols (related to the current capacity of the individual) are preferred to absolute protocols where each individual receives the same amount of training independent of their own capacities [39,40]. A relative training protocol seems to be essential to avoid unintended adverse effects such as reduced EF and reduced mitochondrial respiratory capacity, which have been described with high levels of exercise in animal models [41]. Similarly, populations with highly divergent exercise capacity such as those used in this investigation (LCR and HCR) require individually adapted exercise to avoid over-training in LCR and no training in HCR. Therefore, we tested for exercise capacity according to Hoydal et al. [23] in both selected lines investigated here. We were not able to assess VO_2max_; thus, we used the approximation suggested [23]. We applied high-intensity aerobic interval training, which has been shown to be most efficient with the assessment of each animal, allowing for individually adapted exercise training [40]. Our results indicate proper adaptation for each animal for the following reasons: animals were able to fully perform exercise sessions and could follow the increased intensity (0.02 m/s) of exercise every week. The increase was determined from previous studies conducted by Wang et al. [42] in animals of the same line. Additionally, it has been shown by other groups that the first four weeks of exercise show the most pronounced effect [23,40]. Thus, our protocol used this most effective phase to improve exercise capacity. As a consequence, HCR and LCR improved their best running speed and time to exhaustion (Figure 1), which was more apparent in LCR and both did not show any impairment of cardiac function (Table 1). Exercise training led to an increase in soleus mass and mitochondrial mass, suggesting hypertrophic muscle growth in HCR and LCR. Thus, we can conclude that our exercise training was effective for both HCR and LCR with the latter showing a greater effect of exercise training to improve exercise capacity.

Endurance training has been shown to increase the efficiency of rat skeletal muscle mitochondria, i.e., an increased oxidative phosphorylation efficiency [43]. The response to exercise training with respect to mitochondria in our investigation was twofold. Mitochondrial mass was increased in both HCR and LCR but to a higher percentage in LCR. Mitochondrial respiratory function increased in response to exercise in HCR only. An increase in mitochondrial mass has been related to the volume of exercise training and an increase in mitochondrial activity is positively related to the intensity of exercise [20]. Moreover, Shi et al. found that exercise training in 9- and 18-month-old rats did not affect the metabolic profile in soleus muscle of HCR and LCR [44]. In our investigation, we used young adult rats and found that the volume and intensity of exercise training were similar in both HCR and LCR, but the mitochondrial effects were different. On the one hand, LCR showed a stronger increase in mitochondrial mass following exercise training compared to HCR but on the other hand, only HCR showed an increase in mitochondrial respiratory capacity. These results suggest that the mechanisms by which HCR and LCR improved their exercise capacity are not comparable. With respect to respiratory capacity, we were not able to locate other investigations addressing the response to exercise in a model with differences in intrinsic capacities. Regarding mitochondrial mass, Lessard et al. showed that in soleus muscle, citrate synthase activity was not changed in HCR and LCR after 6 weeks of exercise training [45]. Since they used an incremental protocol with the same absolute cumulative training distance for both strains and thus a different training stimulus compared to our study, it may be one main reason for the different results. Furthermore, the mRNA expression of PGC1α has been described as higher in sedentary LCR compared to HCR [46]. It has also been shown in other models that exercise training increases mitochondrial biogenesis and especially PGC1α expression [47]. Thus, one possible mechanism may be that differences in PGC1α expression are involved in the more pronounced response of LCR mitochondrial mass to exercise training. The effects of exercise training on mitochondrial quantity and mitochondrial quality depend on genetically determined high or low exercise capacity.

It is well known that cardiac IFM and SSM are differentially affected by challenges such as aging, type I or II diabetes, heart failure, or exercise [35]. In skeletal muscle, it has been demonstrated that metabolic pathologies such as obesity and insulin resistance or acute exercise result in greater susceptibility of SSM compared to IFM [14]. In our study, the more pronounced response in mitochondrial function with exercise in HCR compared to LCR was found in IFM but not in SSM. Other investigations distinguishing effects of exercise training between mitochondrial subpopulations are relatively sparse. In the heart, Judge et al. described no changes in IFM and SSM function with lifelong voluntary wheel running but a comparable reduction in hydrogen peroxide production in both mitochondrial subpopulations of the heart [48]. Instead, other training programs such as gradually increasing running times on a treadmill showed more intense proteomic changes in cardiac IFM [49] but also biochemical alterations resisting apoptotic stimuli in both subpopulations [50]. In skeletal muscle, exercise-induced increases in mitochondrial volume [51] or fatty acid oxidation [52,53,54] were described to be more relevant in SSM compared to IFM. A short exercise training program of 2 weeks, however, resulted in similar increases in oxidative capacity in IFM and SSM in quadriceps muscle [55]. Thus, it seems that not only the training program but also skeletal muscle type as well as differences in inherited exercise capacity influence exercise effects on mitochondrial subpopulations. At least in our model of HCR, IFM may play a functional role in mediating exercise effects in skeletal muscle.

Exercise training is suggested to enhance skeletal muscle complex activities differently and maximal respiratory capacity in a substrate-specific manner [53,55]. Respiratory capacity of interfibrillar mitochondria was higher in HCR compared to LCR with essentially most substrates measured, i.e., glutamate/malate, pyruvate malate, and palmitoylcarnitine/malate, and with complex II substrate succinate, complex III substrate DHQ, and complex IV substrate TMPD. Exercise training induced an increase in respiratory capacity in HCR mitochondria with palmitoyl-CoA/malate, palmitoylcarnitine/malate, and complex II substrate succinate and complex III substrate DHQ with only the latter being significant. However, these data indicate that exercise in HCR led to a greater capacity to oxidize fatty acid substrates. In human studies, it has been described that endurance training does affect lipid droplet characteristics [56] and it may augment the intramyocellular lipid content in healthy humans [57]. Intramyocellular lipid droplets are predominantly found at intermyofibrillar regions and it has been suggested that especially IFM functionally interact with these lipid droplets (about 20% of IFM are connected to lipid droplets), enabling efficient ATP production [58]. Furthermore, it has been described that HCR and LCR differ in their properties with respect to substrate oxidation with a gene expression profile of HCR toward more fatty acid oxidation and LCR with a more glycolytic profile [59]. Naples et al. showed that total fatty acid oxidation capacity in red gastrocnemius muscle is higher in HCR compared to LCR [46]. This has been extended with the description of a more efficient use of fatty acids during exercise and an advanced capacity to oxidize fatty acids in HCR [60]. Not only differences in the mitochondrial phenotype but also in the genotype have been described in HCR and LCR [61], which may be causing both basal and exercise-induced differences. Furthermore, during exercise, a higher reliance on fat oxidation in proportion to carbohydrates has been described [62,63]. Accordingly, ß-oxidation and fatty acid transport can be increased through the regulation of fatty acid transporters with exercise training [64,65,66]. Thus, our findings of an increase in fatty-acid-related respiratory capacity in HCR‘s IFM with exercise training seem reasonable.

One of the described effects of exercise training on mitochondria is an increase in isolated complex activities. In mice, complex I, II, III, and IV activity was significantly increased following exercise training and correlated with the frequency [67]. Mitochondrial complex activities are related to mitochondrial respiratory capacity. In our investigation, exercise led to an increase in complex activities with IFM in HCR but not in LCR. These results correlate well with the change in respiratory capacity following exercise in both HCR and LCR. Thus, respiratory capacity in HCR‘s IFM may be partly increased due to the increased function of the separate mitochondrial complexes following exercise. Contrary to the findings in IFM, SSM showed increased activities in almost all complexes in HCR and LCR, which was not reflected by respiratory capacity, which remained unchanged. The respiratory chain complexes form supercomplexes (mainly consisting of complex I, III, IV), enabling an increased stability and leading to higher activity. Therefore, increases in supercomplexes may increase mitochondrial respiration. In human skeletal muscle, increases in supercomplexes’ assembly have been observed in parallel with muscle respiration [68]. One potential reason for the discrepancy between increased isolated complex activities of SSM in HCR and LCR and no changes in respiration may be a limitation in the formation of supercomplexes, which was, however, not investigated in our analysis. We show for the first time that the effect of interval training was dependent on genetically determined intrinsic exercise capacity as well as mitochondrial subpopulations.

Our high-intensity exercise training led to cardiac hypertrophy and increased skeletal muscle mitochondrial respiratory capacity in HCR only. In contrast, running speed, time to exhaustion, as well as skeletal muscle mitochondrial mass were increased more pronouncedly in LCR. Additionally, exercise led to an increase in glucose tolerance in LCR but not in HCR. Thus, the improvement in exercise capacity in LCR may be due to mechanisms such as higher (pulmonary) respiratory capacity, neurohumoral activity, or cardiovascular adaptations. These findings further indicate that several effects of exercise depend on intrinsic exercise capacity. In male rats, it has been shown by us that exercise training affects deoxyglucose uptake and citrate synthase activity in several tissues in HCR but not in LCR [25]. Similarly, results from investigations in humans indicate that exercise training in genetically predisposed populations seems to be frequently ineffective as has been shown for African, Arabic, Chinese, or Polynesian subjects [69]. Thus, intrinsic exercise capacity (as well as its combination with sex) appears to be important not only for total exercise capacity but also for the response to exercise training. It seems that improvement in mitochondrial quantity is one determining factor for improving exercise capacity in HCR and LCR. Altogether, LCR showed a greater response to 4 weeks of high-intensity exercise, indicating higher trainability.

### 4.1. Limitations

Mitochondrial networks are formed and potential mito-organelle interactions are discussed as important for the overall mitochondrial function. Here, we used isolated mitochondria where these possible interactions cannot be assessed. Instead, our intention was to determine differences in mitochondrial function in interfibrillar and subsarcolemmal mitochondria separately. In our investigation, we used female rats. It has been shown that the menstrual phase in females has no influence on physical performance [70]. Since we used several animals per group and therefore different cycle phases are represented, we should be able to rule out the influence of sex hormones on our experiments. Nevertheless, it would be important to investigate sex differences in the future. Moreover, our investigation does not allow for predictions on different muscle fiber types or conclusions on any other skeletal muscle except for gastrocnemius. Importantly, gastrocnemius muscle significantly contributes to propulsive strength when running and is primarily involved in running and jumping and was thus selected for this investigation. 

### 4.2. Conclusions

High intrinsic exercise capacity is associated with increased mitochondrial function in gastrocnemius muscle interfibrillar mitochondria compared to low intrinsic exercise capacity. This suggests that IFM in particular play a role in the difference between the skeletal muscle capacity of HCR and LCR. Thus, differences in skeletal muscle mitochondrial subpopulations seem to be mainly inherited. Moreover, four weeks of high-intensity interval training was more effective in LCR, showing a higher trainability under these conditions. Interestingly, HCR primarily adapted after exercise training with higher IFM respiratory capacity and thus higher mitochondrial quality, while LCR significantly increased mitochondrial mass and thus an increase in mitochondrial quantity. Therefore, genetic predisposition seems to have a strong effect on how exercise training affects the rat organism and metabolism. Our results suggest that the mechanisms of skeletal muscle mitochondrial response to exercise training differ in dependence on inherited high or low intrinsic exercise capacity. We speculate that an increase in exercise capacity may be achieved solely by increasing mitochondrial mass and/or function. Furthermore, the response to exercise in humans can be expected to be mainly dependent on genetic factors.

## Figures and Tables

**Figure 1 cells-13-00393-f001:**
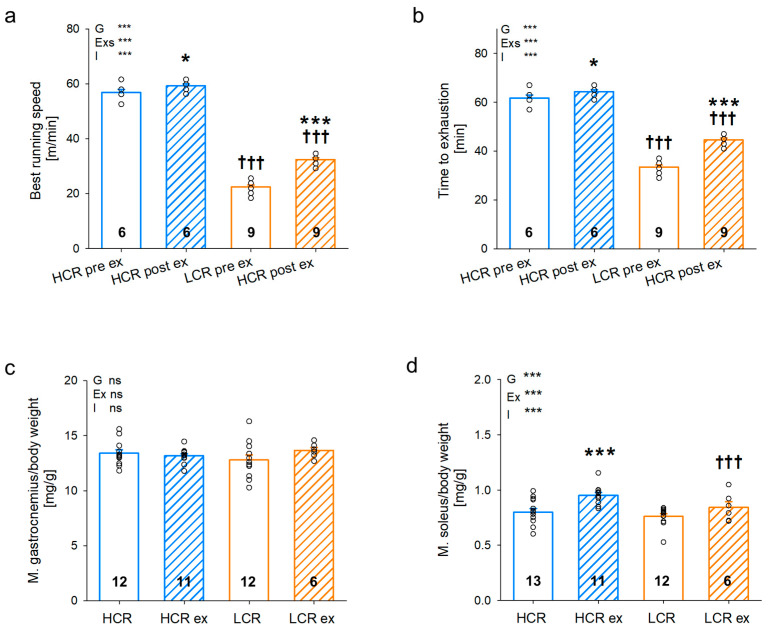
Treadmill exercise test pre and post exercise training (**a**,**b**) and muscle weights (**c**,**d**) of rats with inherited high (HCR) or low (LCR) exercise capacity with and without exercise training; data are mean ± SEM, blue—HCR, orange—LCR, G—inherited exercise capacity, Exs—exercise status pre or post exercise, Ex—exercise training, I—interaction, ns—non-significant, n = 6–9 for exercise test, n = 6–13 for muscle weights, * *p* < 0.05 and *** *p* < 0.001 for G, Exs, Ex, I, or compared tothe same group without exercise training, ††† *p* < 0.001 compared to HCR with the same treatment; for exercise test, two-way repeated measurement ANOVA was performed and for other experiments, two-way ANOVA was performed.

**Figure 2 cells-13-00393-f002:**
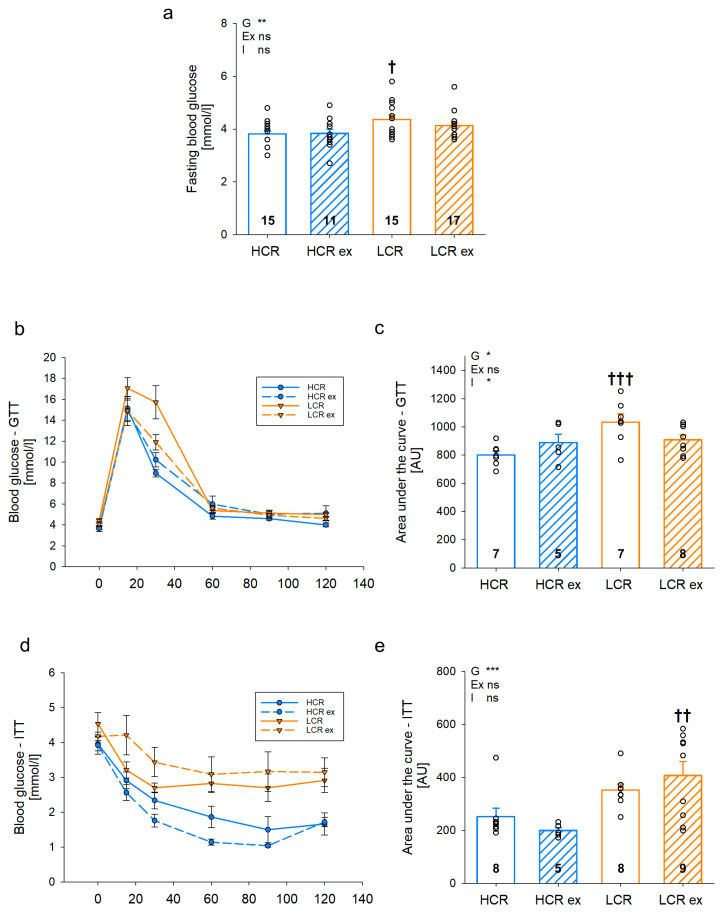
Fasting blood glucose (**a**), glucose tolerance (GTT; (**b**,**c**)), and insulin tolerance (ITT; (**d**,**e**)) test of rats with inherited high (HCR) or low (LCR) exercise capacity with and without exercise training; data are mean ± SEM, blue—HCR, orange—LCR, G—inherited exercise capacity, Ex—exercise training, I—interaction, ns—non-significant, n = 11–17 for fasting blood glucose, n = 5–9 for GTT and ITT, * *p* < 0.05, ** *p* < 0.01, and *** *p* < 0.001 for G, Ex, I, or compared tothe same group without exercise training, † *p* < 0.05, †† *p* < 0.01, and ††† *p* < 0.001 compared to HCR with the same treatment. Two-way ANOVA was performed.

**Figure 3 cells-13-00393-f003:**
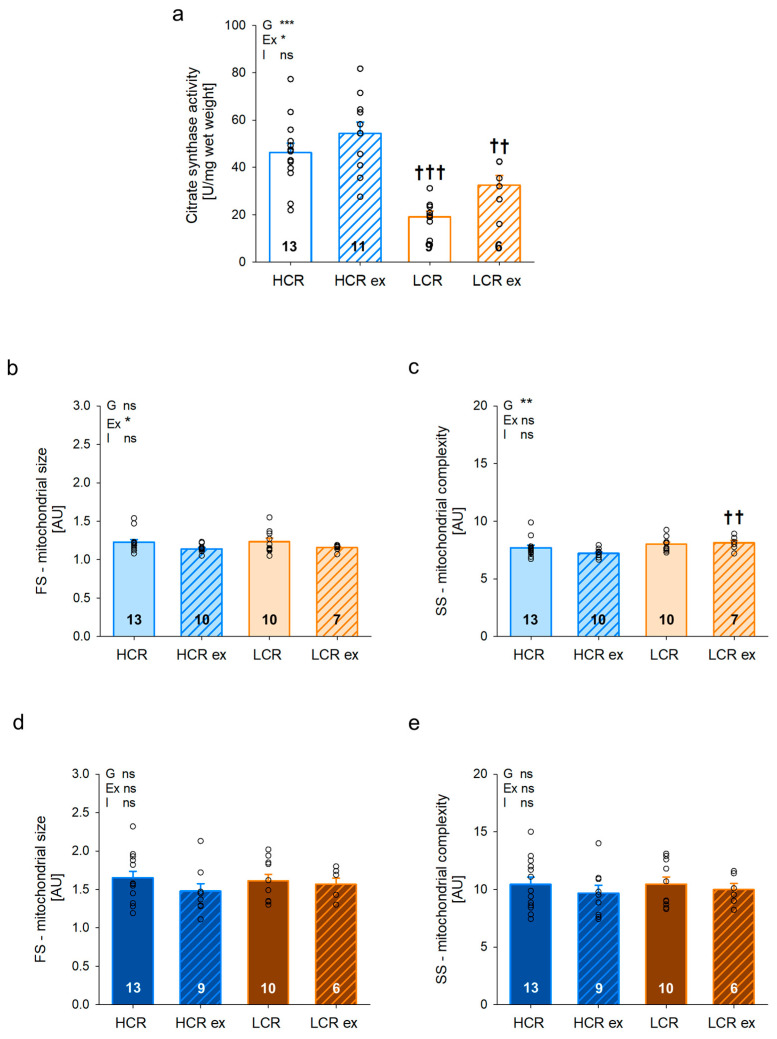
Citrate synthase activity (**a**), size (**b**,**d**), and complexity (**c**,**e**) of interfibrillar (IFM; (**b**,**c**)) and subsarcolemmal (SSM; (**d**,**e**)) mitochondria from *Musculus gastrocnemius* of rats with inherited high (HCR) or low (LCR) exercise capacity with and without exercise training; data are mean ± SEM, blue—HCR, orange—LCR, light color—IFM, dark color—SSM, FS—forward scatter, SS—sideward scatter, G—inherited exercise capacity, Ex—exercise training, I—interaction, ns—non-significant, n = 6–13, * *p* < 0.05, ** *p* < 0.01, and *** *p* < 0.001 for G, Ex, I, or compared tothe same group without exercise training, †† *p* < 0.01 and ††† *p* < 0.001 compared to HCR with the same treatment. Two-way ANOVA was performed.

**Figure 4 cells-13-00393-f004:**
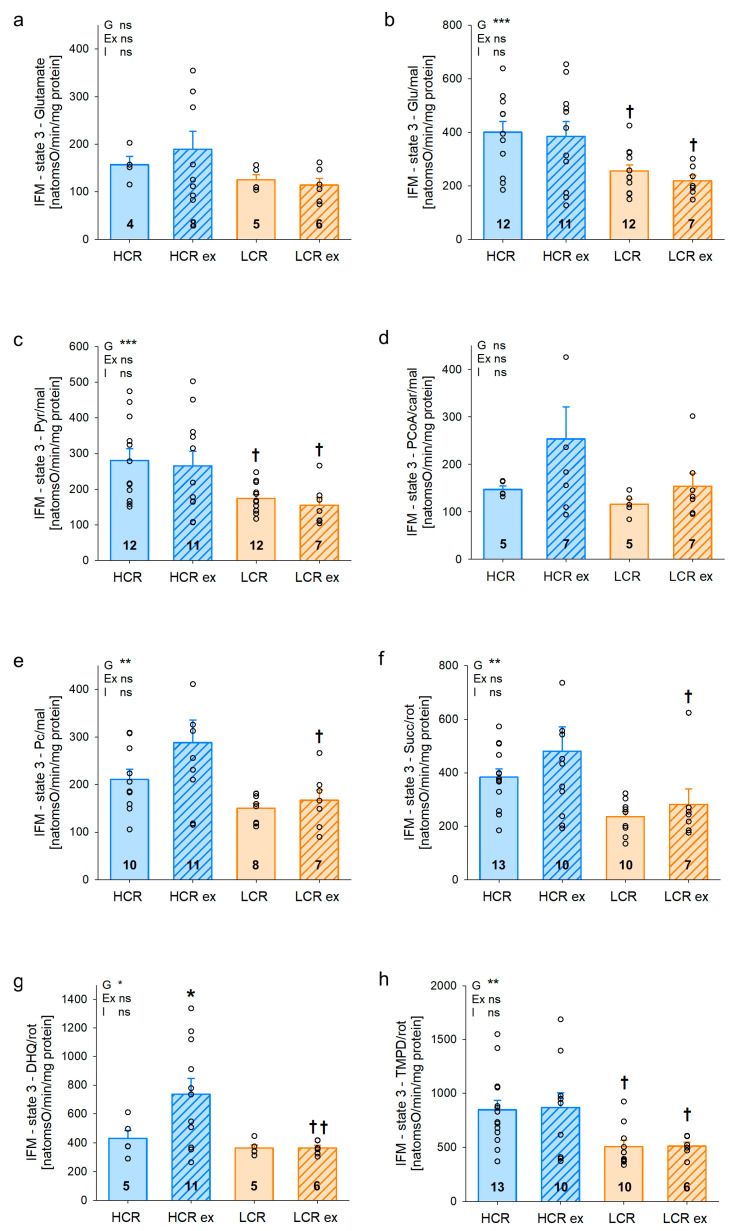
Maximal respiratory capacity of interfibrillar (IFM) mitochondria from *Musculus gastrocnemius* of rats with inherited high (HCR) or low (LCR) exercise capacity with and without exercise training using different substrates (**a**–**h**); data are mean ± SEM, blue—HCR, orange—LCR, G—inherited exercise capacity, Ex—exercise training, I—interaction, ns—non-significant, Glu—glutamate, mal—malate, pyr—pyruvate, PCoA—palmitoyl coenzyme A, car—carnitine, pc—palmitoylcarnitine, succ—succinate, rot—rotenone, DHQ—durohydroquinone, TMPD—N,N,N′,N′-Tetramethyl-p-phenylendiamine, n = 8–14, * *p* < 0.05, ** *p* < 0.01, and *** *p* < 0.001 for G, Ex, I, or compared tothe same group without exercise training, † *p* < 0.05 and †† *p* < 0.01 compared to HCR with the same treatment. Two-way ANOVA was performed.

**Figure 5 cells-13-00393-f005:**
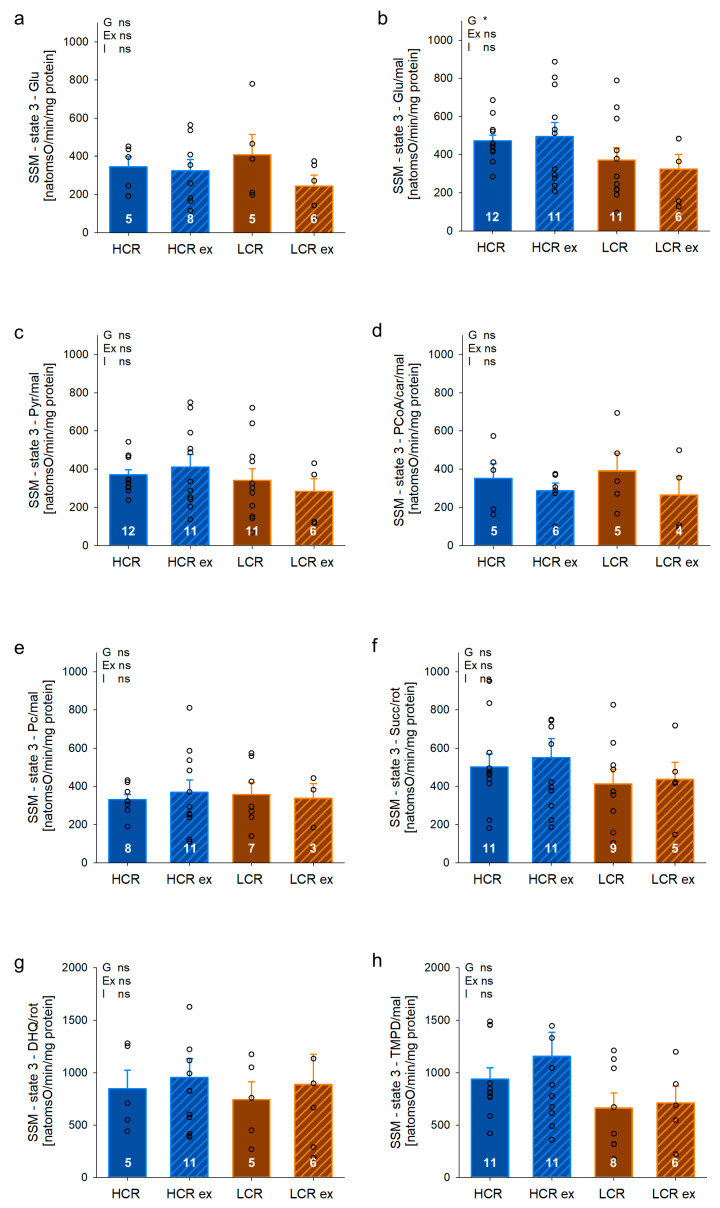
Maximal respiratory capacity of subsarcolemmal (SSM) mitochondria from *Musculus gastrocnemius* of rats with inherited high (HCR) or low (LCR) exercise capacity with and without exercise training using different substrates (**a**–**h**); data are mean ± SEM, blue—HCR, orange—LCR, G—inherited exercise capacity, Ex—exercise training, I—interaction, ns—non-significant, Glu—glutamate, mal—malate, pyr—pyruvate, PCoA—palmitoyl coenzyme A, car—carnitine, pc—palmitoylcarnitine, succ—succinate, rot—rotenone, DHQ—durohydroquinone, TMPD—N,N,N′,N′-Tetramethyl-p-phenylendiamine, n = 3–12, * *p* < 0.05 for G. Two-way ANOVA was performed.

**Figure 6 cells-13-00393-f006:**
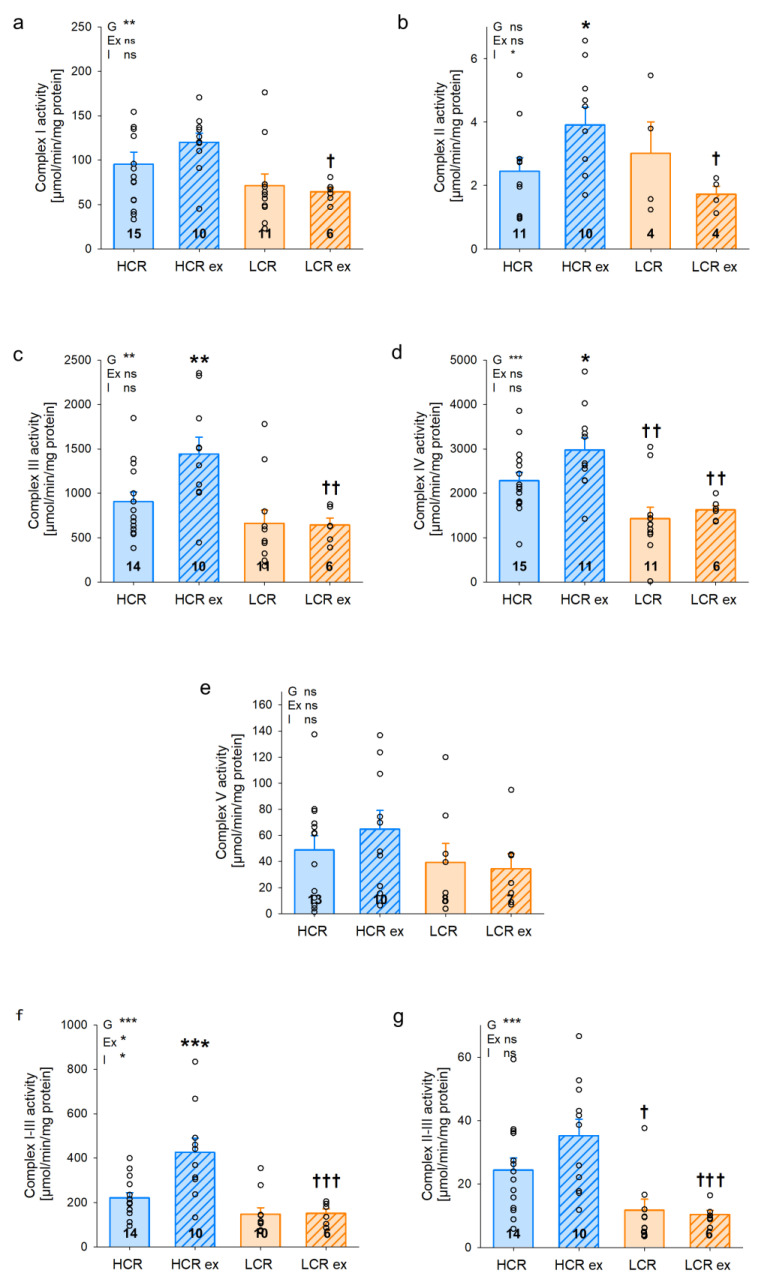
Isolated complex activities and complex combinations of interfibrillar (IFM; (**a**–**g**)) mitochondria from *Musculus gastrocnemius* of rats with inherited high (HCR) or low (LCR) exercise capacity with and without exercise training; data are mean ± SEM, blue—HCR, orange—LCR, G—inherited exercise capacity, Ex—exercise training, I—interaction, ns—non-significant, n = 4–15, * *p* < 0.05, ** *p* < 0.01, and *** *p* < 0.001 for G, Ex, I, or compared tothe same group without exercise training, † *p* < 0.05, †† *p* < 0.01, and ††† *p* < 0.001 compared to HCR with the same treatment. Two-way ANOVA was performed.

**Figure 7 cells-13-00393-f007:**
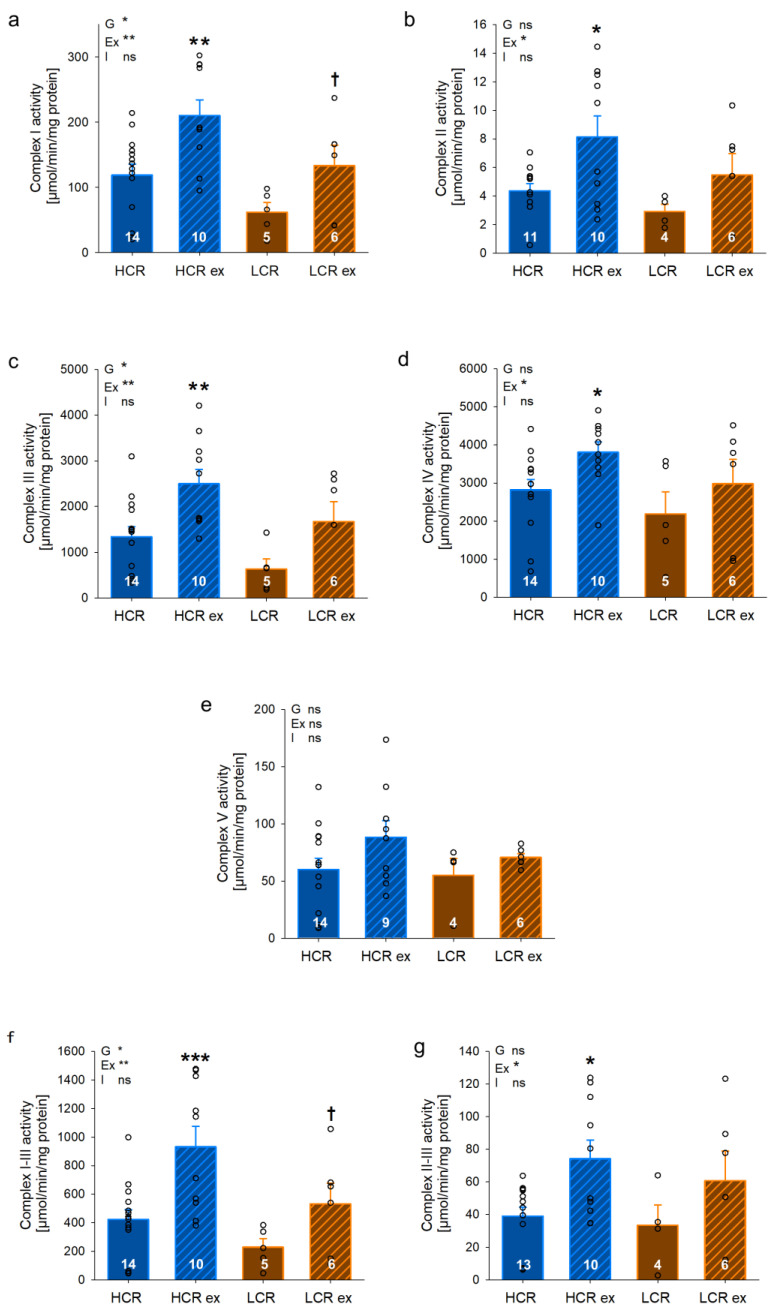
Isolated complex activities and complex combinations of subsarcolemmal (SSM; (**a**–**g**)) mitochondria from *Musculus gastrocnemius* of rats with inherited high (HCR) or low (LCR) exercise capacity with and without exercise training; data are mean ± SEM, blue—HCR, orange—LCR, G—inherited exercise capacity, Ex—exercise training, I—interaction, ns—non-significant, n = 4–14, * *p* < 0.05, ** *p* < 0.01, and *** *p* < 0.001 for G, Ex, I, or compared tothe same group without exercise training, † *p* < 0.05 compared to HCR with the same treatment. Two-way ANOVA was performed.

**Table 1 cells-13-00393-t001:** Morphometry and echocardiographic parameters of sedentary and trained rats with high (HCR) or low (LCR) intrinsic exercise capacity.

	HCR (n)	HCR ex (n)	LCR (n)	LCR ex (n)	G	E	I
Age [weeks]	15.1 ± 0.3 (15)	15.6 ± 0.1 (11)	15.3 ± 0.2 (12)	15.4 ± 0.3 (7)	ns	ns	ns
Body weight [g]	177 ± 3 (14)	204 ± 8 ** (11)	213 ± 6 ††† (12)	204 ± 8 (6)	**	ns	**
Tibia length (TL) [mm]	34.1 ± 0.4 (15)	34.5 ± 0.2 (11)	34.8 ± 0.4 (12)	35.1 ± 0.4 (6)	ns	ns	ns
Heart weight/TL [mg/mm]	17.2 ± 0.3 (15)	20.3 ± 0.6 *** (11)	17.4 ± 0.6 (12)	18.2 ± 0.6 † (6)	ns	***	*
Lung weight/TL [mg/mm]	24.9 ± 0.4 (15)	27.9 ± 0.7 ** (10)	25.9 ± 0.7 (12)	26.2 ± 1.2 (6)	ns	*	ns
Liver weight/TL [g/mm]	181 ± 3 (15)	233 ± 10 *** (11)	211 ± 10 †† (12)	208 ± 11 (6)	ns	**	**
Heart rate [beats/min]	380 ± 14 (8)	339 ± 7 * (6)	384 ± 11 (8)	370 ± 12 (10)	ns	*	ns
IVSd + LVPWd [mm]	3.14 ± 0.11 (8)	3.25 ± 0.18 (6)	3.34 ± 0.11 (8)	3.37 ± 0.09 (10)	ns	ns	ns
LVIDd [mm]	6.41 ± 0.16 (8)	6.92 ± 0.13 (6)	6.10 ± 0.19 (8)	6.42 ± 0.16 † (10)	*	*	ns
E/E’	14.2 ± 0.6 (8)	15.1 ± 0.6 (6)	14.4 ± 1.0 (8)	17.5 ± 1.1 (10)	ns	ns	ns
FS [%]	41.8 ± 2.1 (8)	41.6 ± 2.9 (6)	53.6 ± 2.6 ††† (8)	47.6 ± 2.1 (10)	***	ns	ns

Data are mean ± SEM. Ex—exercise-trained, G—genetically determined aerobic exercise capacity, E—exercise training, I—interaction, IVSd—anterior wall thickness in diastole, LVPWd—left ventricular posterior wall thickness in diastole, LVIDd—left ventricular internal dimension in diastole, E/E‘—ratio between early mitral inflow velocity and mitral annular early diastolic velocity, FS—fractional shortening, ns—non-significant, n = 6–15 for morphometry, n = 6–10 for echocardiography, * *p* < 0.05, ** *p* < 0.01, and *** *p* < 0.001 for G, E, I, or compared to sedentary animals † *p* < 0.05, †† *p* < 0.01, and ††† *p* < 0.001 in comparison to HCR with the same treatment. Two-way ANOVA was performed.

## Data Availability

Data are available from the corresponding author by reasonable request.

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
