# Peer review of "Exercise Training Differentially Affects Skeletal Muscle Mitochondria in Rats with Inherited High or Low Exercise Capacity"

_cells, 2024, doi:10.3390/cells13050393_

Round 1

Reviewer 1 Report

Comments and Suggestions for Authors

Dear Authors,

Manuscript Number: Cells-2828720

Title Manuscript: Inherited exercise capacity differentially affects the response of skeletal muscle mitochondria to exercise training

This study is an interesting topic but at the moment MAJOR REVISIONS are necessary in order to make it suitable for a final decision for “Cells”.

Summary statement: This experimental study examined the Effect of 4 weeks of high-intensity intervals on skeletal muscle mitochondrial function (gastrocnemius), echocardiographic parameters, aerobic exercise capacity, glucose and insulin tolerance test, as well as mitochondrial indices in female rat models (High/HCR and low/LCR). The results revealed that exercise training increased exercise capacity in both HCR and LCR female rat models, but this was more pronounced in LCR. In addition, exercise increased skeletal muscle mitochondrial mass more in LCR. Instead, maximal respiratory capacity was increased following exercise in HCRs interfibrillar mitochondria (IFM) only.  

POINTs of STRENGTH:

1) Effect of exercise training on skeletal muscle mitochondrial function in rat models with divergent inherited exercise capacity;

2) Arguments in the discussion section;

POINTs of WEAKNESS (and/or should be revised to improve the manuscript):

Main title

3) The type of participants (rat models) in the main title is not specified. Please specify;

Abstract:

4) The type of study, total rats, gender of rats, groups and their number in each group, mean age and weight of rats, and training protocol are not specified in the methods section of the abstract. Please provide;

5) The significance level of the results is not specified. Please specify In the results section of the abstract;

1. Introduction:

6) The hypothesis (s) of this study is not specified; please specify clearly;

2. Materials and methods

Animal

7) Please specify the total rats and number of rats in each group in the animal model section;

Aerobic interval training

8) Please specify the time of training protocol (morning or other time, and hour);

Statistical analysis

9) Did authors use a statistical software to calculate the sample size? IF YES, please explain and add its name and valid reference in the statistical analysis section.

10) The significance level of statistical analysis was considered for two-tailed OR one-tailed…? Please specify;

3. Results

11) Please specify the significance level for the results in the results section of the manuscript;

12) IF possible, please provide other echocardiographic parameters such as (Left ventricular end‑diastolic diameter, Left ventricular end‑systolic diameter, Left ventricular septum diastolic diameter, Left ventricular posterior wall diastolic diameter, Aortic root diameter, Left atrial area diameter, Right ventricular diameter, Right atrial area volume, E velocity, A velocity, Mitral E-wave deceleration time, Aortic velocity time integral, Pulmonary velocity time integral, Left ventricular ejection fraction, End-diastolic volume, End-systolic volume, Cardiac output, and Cardiac index) in the Table 1 and/or in this manuscript;

13) Please provide the number of rats for each group in the Tables 1-3;

4. Discussion &  Conclusions

14) What are the conclusions and implications for future research?;

15) What does this experimental study add to the literature?;

References

16) References section is not always in accordance with authors' guidelines. In particular, please check No. 4, 6, 14, 19, 26, 38, 45, 53, and 63 for validation.

Best Regards

January 8, 2024

Author Response

We thank the reviewers and the editor for their constructive comments. We are pleased to read that the manuscript seems potentially acceptable. We have revised the entire manuscript to address the comments of the three reviewers. We have taken special care revising the content of the discussion to highlight the novelty of our results of the study. Please find below our point-by-point response.

Reviewer 2 Report

Comments and Suggestions for Authors

Thank you for the opportunity to review this manuscript. The paper is highly interesting and makes a significant contribution to understanding the function of mitochondrial subpopulations in skeletal muscles with high or low inherited exercise capacities. Additionally, the authors explore the effects of a four-week high-intensity training program on the skeletal muscles of rats using a treadmill. The authors suggest, among other things, that the mechanisms of skeletal muscle mitochondrial response to exercise training differ depending on the inherited high or low intrinsic exercise capacity. The manuscript presents valuable insights into the impact of a 4-week high-intensity training on mitochondrial subpopulations in skeletal muscles with high or low inherited exercise capacity.

The authors have prepared an acceptable manuscript. However, I have concerns regarding the existing issues in the manuscript.

Major Issues:

Firstly, I have serious objections regarding the number of animals used in individual groups and, consequently, the research results described and presented.

On line 131, it was mentioned that HCR and LCR rats at the age of ~11 weeks were assigned to one of four groups (n = 10/group). According to this information, there should be four groups with 10 rats in each. However, there are discrepancies in the legends of the figures:

             Fig. 1 legend, line 218: n=6-9 for exercise; n=6-13 for muscle weights; n=5-9 for GTT and ITT

             Fig. 2 legend, line 276: n=6-13

             Fig. 3 legend, line 310: n=8-14

             Fig. 4 legend, line 342: n=3-11

             Fig. 5 legend, line 375: n=4-15

Furthermore, I tried to count the dots in individual bars in the figures and I think that the inconsistencies are also observed. Similar discrepancies related to 'n' per each group are also present in the tables.

Moreover, the methods are not well described. For example, the CS activity assay should have been measured in mitochondria (line 150), but it was not. Additionally, both activities should be expressed as U/mg protein.

Regarding the isolation of mitochondria (line 145, ref. 16), I checked this paper and found a reference to another paper (Biochemical properties of subsarcolemmal and interfibrillar mitochondria isolated from rat cardiac muscle; Palmer et al., JBC, 1977). Why do the authors cite themselves, if they did not make any modifications in the method, please explain. I also suggest using a more specific method to isolate mitochondria from skeletal muscle, for example (Isolation of mitochondrial subpopulations from skeletal muscle: optimizing recovery and preserving integrity, Lai et al., Acta Physiol, 2019).

Considering the significant differences in the 'n' for individual measurements without any explanation from the authors and the methods used, it is challenging to assume that the discussion and conclusions drawn are correct.

Comments on the Quality of English Language

The quality of the English Language is fine.

Author Response

(The authors gave the same response as above.)

Reviewer 3 Report

Comments and Suggestions for Authors

Heyne and colleagues assessed changes in interfibrillar and subsarcollemal mitochondrial population function following exercise training in rats with high (HCR) and low (LCR) inherited running capacity. The authors obtained important results that allow to evaluate the contribution of two mitochondrial populations to adaptation to physical stress. I had several comments about the work:

Comments:  

1. What strain of rats was the original one for obtaining HCR and LCR animals?

2. It is known that even in the Wistar rat line there are animals with high resistance and low resistance to acute hypoxia. The tissue's response to physical stress is also determined by its resistance to the development of hypoxic conditions. I encourage the authors to discuss their results based on possible animal differences in hypoxia tolerance.

2. Why did the authors use females? This needs to be justified in the text. Most mitochondrial studies use males due to the possible influence of the menstrual cycle on mitochondrial function.

3. The results obtained should be discussed in the context of the different roles of the IFM and SSM fractions of mitochondria in skeletal muscle. It is known that the SSM fraction takes an active part in calcium homeostasis. While the IFM population is more involved in energy supply to the tissue, as evidenced by the authors’ results.

4. For a better understanding Fig. 1 should be divided and presented separately panels A-D, panels E-I should be presented in the form of Fig. 2. This will increase the size of the figures; in the presented form they are very inconvenient to read. The size of figures 2, 3, 4 and 5 should also be increased; arrange the drawings in several rows. I propose that some of the figures and tables (for example, tables 2 and 3) be transferred to supplementary materials.

5. In Materials and Methods, the authors describe the isolation of interfibrillar mitochondria, but the results also provide results for subsarcollemal mitochondria. In the materials and methods section, the methods for isolating the two indicated fractions of muscle mitochondria should be briefly described.

6. The authors need to clearly explain to the reader the meaning of the term mitochondrial complexity.

Author Response

(The authors gave the same response as above.)

Round 2

Reviewer 1 Report

Comments and Suggestions for Authors

Dear Authors,

Manuscript Number: Cells-2828720

Title Manuscript: Exercise training differentially affects skeletal muscle mitochondria in rats with inherited high or low exercise capacity

I am very grateful to the authors for their efforts.

In general, although this manuscript has found good content after correcting major revisions, some concerns and/or MINOR REVISIONs have to be addressed before a final version can be made:

POINTs of WEAKNESS (and/or should be revised to improve the manuscript):

Abstract:

1) The type of study, total rats, number of groups, number of rats in each group, and mean weight of rats are not specified in the methods section of the abstract. Please provide;

2) The significance level of the results is not specified. Please specify in the results section of the abstract;

 2. Materials and methods

Statistical analysis

3) The significance level of statistical analysis was considered for two-tailed OR one-tailed…? Please specify;

3. Results

4) Please specify the significance level for the results in the results section of the manuscript;

References

5) References section is not always in accordance with authors' guidelines. In particular, please check No. 14 and 19 for validation.

Best Regards

January 29, 2024

Author Response

We thank the reviewer for his efforts to improve our manuscript.

Reviewer 2 Report

Comments and Suggestions for Authors

I apologize, but I cannot accept your explanation regarding the number of animals used per group. I understand that there can be lower measurements at the beginning of the experiment due to various factors, as you wrote in your Response to reviewers. However, in my opinion, it is not acceptable, for such big differences in the initial phase of the experiment for example, in Table 1 “n” value for:

HCR       HCRexe               LCR        LCRexe

age                                      n=14     n=11                    n=12     n=7

Heart rate [beats/min]   n=8        n=6                       n=8        n=10

For the figures, there is the same higher n than at the beginning of the experiment, for example:

“Animals were euthanized, and organs and tissues harvested at ~ 15 weeks of age (HCR n=14,

HCR ex n=14, LCR n=12, LCR ex n=8)”.

Fig. 3 CS activity was measured in homogenate

HCR       HCRexe               LCR        LCRexe

n=13      n=11                     n=9        n=6

Then, homogenate must be used to isolate the mitochondrial fraction. Even though I understand the lower number of mitochondria due to your explanation, I do not have any rational explanation for the higher number of mitochondria.

Fig. 6 a, d, h

HCR       HCRexe               LCR        LCRexe

n=15      n=10                     n=11      n=6 (a)

n=15      n=11                     n=11      n=6 (d)

n=14      n=11                     n=10      n=7 (h)

Fig. 7 a

n=15      n=11                     n=11      n=7 (a)

Another confusing aspect is the significant differences in the number of measurements using a Clark-type oxygen electrode. Meanwhile, I may accept a lower number of measurements (±2, between groups) from my perspective and research experience, differences in “n” 5 or even 6 between groups are not acceptable. Please take a look, for example, at Fig. 5.

Author Response

(The authors gave the same response as above.)

Reviewer 3 Report

Comments and Suggestions for Authors

The authors adequately responded to my comments and concerns.

Author Response

(The authors gave the same response as above.)

Round 3

Reviewer 1 Report

Comments and Suggestions for Authors

Dear Authors,

Manuscript Number: Cells-2828720

Title Manuscript: Exercise training differentially affects skeletal muscle mitochondria in rats with inherited high or low exercise capacity

I am very grateful to the authors for their efforts.

In general, this manuscript has found suitable content after correcting revisions, and the modified revisions are accepted.  

Best Regards

Feb 14, 2024

Author Response

We thank for the comments and revised the manuscript accordingly.
